# Parameter-efficient Tuning for Large Language Model without Calculating Its Gradients

**Feihu Jin**[1,2], **Jiajun Zhang**[1,2,3,4] * and **Chengqing Zong**[1,2]

[1]Institute of Automation, Chinese Academy of Sciences, Beijing, China
[2]School of Artificial Intelligence, University of Chinese Academy of Sciences, Beijing, China
[3]Wuhan AI Research
[4]Shanghai Artificial Intelligence Laboratory, Shanghai, China
jinfeihu2020@ia.ac.cn,[†]{jjzhang,cqzong}@nlpr.ia.ac.cn

## Abstract

Fine-tuning all parameters of large language models (LLMs) requires significant computational resources and is time-consuming. Recent parameter-efficient tuning methods such as Adapter tuning, Prefix tuning, and LoRA allow updating a small subset of parameters in large language models. However, they can only save approximately 30% of the training memory requirements because gradient computation and backpropagation are still necessary for these methods. This paper proposes a novel parameter-efficient tuning method for LLMs without calculating their gradients. Leveraging the discernible similarities between the parameter-efficient modules of the same task learned by both large and small language models, we put forward a strategy for transferring the parameter-efficient modules derived initially from small language models to much larger ones. To ensure a smooth and effective adaptation process, we introduce a Bridge model to guarantee dimensional consistency while stimulating a dynamic interaction between the models. We demonstrate the effectiveness of our method using the T5 and GPT-2 series of language models on the SuperGLUE benchmark. Our method achieves comparable performance to fine-tuning and parameter-efficient tuning on large language models without needing gradient-based optimization. Additionally, our method achieves up to 5.7× memory reduction compared to parameter-efficient tuning.

## 1 Introduction

Large language models such as GPT3 (Brown et al., 2020), GPT4 (OpenAI, 2023), T5-XXL (Raffel et al., 2020), and LLaMA (Touvron et al., 2023) have demonstrated remarkable capabilities in various natural language processing tasks. However,

---

*Corresponding Author
† The author is currently a PhD student in Peking University. The new email address is fhjin@stu.pku.edu.cn

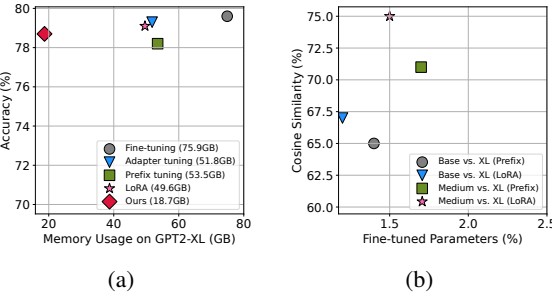

Figure 1: (a) Comparison of the GPU memory usage during training for fine-tuning, Adapter tuning, LoRA, Prefix tuning, and our method on the RTE task. (b) The cosine similarity of the parameter-efficient modules on the GPT-2 series of language models over the RTE task.

the sheer number of parameters in these models poses challenges for fine-tuning on common hardware. Parameter-efficient tuning methods (e.g., Adapter tuning, Prefix tuning, or LoRA) typically involve adding a small number of parameters to the language model and only fine-tuning the added subset of parameters, achieving comparable performance to full fine-tuning. Adapter tuning learns the task-specific information (Houlsby et al., 2019; Mahabadi et al., 2021b,a) by inserting small task-specific modules within layers of the Transformer. Prefix tuning (Li and Liang, 2021; Liu et al., 2021b) prepends task-specific trainable prompt tokens to the hidden states within every intermediate Transformer layer. LoRA (Hu et al., 2021) merges the low-rank and trainable matrices with the frozen weights at each layer of the Transformer. In Figure 1(a), we show the comparison of the GPU memory usage during training for fine-tuning, Adapter tuning, LoRA, and Prefix tuning on the Recognizing Textual Entailment (RTE) (Bar-Haim et al., 2014) task. These parameter-efficient tuning methods can save approximately 30% of the GPU memory requirements but still rely on gradient-based optimization, resulting in increased memory demands for LLMs.

To address the above limitations, we propose a novel parameter-efficient tuning method for LLMs without calculating their gradients. Intuitively, both large and small language models (SLMs) can learn similar task-specific characteristics when applied to downstream tasks. We conduct experiments using existing parameter-efficient tuning methods on the RTE task in the SuperGLUE benchmark (Wang et al., 2019) to validate this hypothesis. As depicted in Figure 1(b), we calculate the similarity between the parameter-efficient tuning modules derived from GPT2-XL and GPT2-base or GPT2-medium[1]. Specifically, we apply the LoRA method on GPT2-XL and GPT2-medium to obtain the parameter-efficient modules of the two language models and calculate the cosine similarity of the two parameter-efficient modules. We find that the cosine similarity can reach up to 75%. Our observations indicate that these modules exhibit comparable task-specific characteristics throughout the learning process for specific downstream tasks. Inspired by these findings, if we can successfully transfer the task-specific characteristics learned by the small language model to the large language model, we can enrich the task-specific capabilities into the LLMs without needing gradient-based optimization.

In this paper, we first utilize existing parameter-efficient tuning methods in a small language model to learn the task characteristics of downstream tasks. Intuitively, we can directly apply the parameter-efficient module obtained from the small language model to the large language model. However, this would face crucial issues of dimension mismatch and limited interaction with the large language model. To address the issue of dimension mismatch, we employ a projection module to align the dimensions of the parameter-efficient modules between SLMs and LLMs. Furthermore, to enrich the interaction between the parameter-efficient module and the large language model, we introduce a Bridge model that can retain the knowledge of the large language model while interacting with the parameter-efficient module, obtaining a parameter-efficient module with dimensions matching the large language model. Finally, we seamlessly plug the acquired parameter-efficient module into the large language model for inference. We conduct comprehensive experiments on T5 series

---

[1]GPT2-base contains 117M parameters, GPT2-medium contains 345M parameters, and GPT2-XL contains 1542M parameters.

(Raffel et al., 2020) and GPT2 series (Radford et al., 2019) of language models to assess the effectiveness of our method using the SuperGLUE benchmark, a widely recognized evaluation benchmark for natural language understanding. The results demonstrate that our method performs on par with fine-tuning and parameter-efficient tuning on large language models without needing gradient-based optimization. Additionally, our proposed method achieves up to $5.7\times$ memory reduction compared to parameter-efficient tuning. Our findings highlight the potential of bridging small and large language models, thereby efficiently leveraging expansive large language models. In summary, our key contributions can be listed as follows:

- Our analysis reveals a substantial task similarity when applying parameter-efficient tuning methods to SLMs and LLMs for downstream tasks.

- We propose a gradient-free method to adapt the parameter-efficient modules learned in a small language model to a large language model.

- Extensive experiments on SuperGLUE benchmark under both T5 series and GPT-2 series of language models verify the effectiveness of our proposed method and achieve up to $5.7\times$ memory reduction compared to parameter-efficient tuning.

## 2 Method

The proposed method utilizes parameter-efficient tuning modules to effectively bridge small and large language models, enriching the task-specific capabilities into the large language model without needing gradient-based optimization. As depicted in Figure 2, the method consists of training and inference. We first employ parameter-efficient tuning methods during the training stage to learn task-specific characteristics in the small language model. Then, we fine-tune the Bridge model and the acquired parameter-efficient module to enhance the knowledge of the parameter-efficient module. Finally, we directly plug the parameter-efficient module into the large language model during the inference stage for efficient predictions.

**Plug-in and Bridge Model Fine-tuning:** We first utilize existing parameter-efficient tuning methods such as Adapter (Houlsby et al., 2019), LoRA (Hu

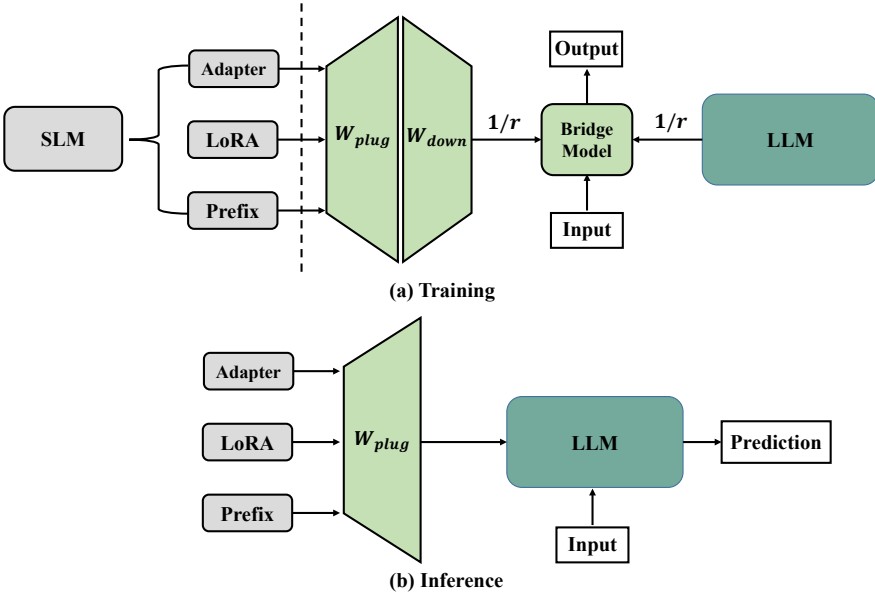

**(a) Training**

**(b) Inference**

Figure 2: (a) In the training stage, we use parameter-efficient tuning methods to learn task-specific characteristics in a smaller model and fine-tune the Bridge model with the acquired parameter-efficient modules. (b) In the inference stage, we directly plug the parameter-efficient modules into the large language model for efficient predictions.

et al., 2021), or Prefix tuning (Li and Liang, 2021) in a small language model to learn the task characteristics of downstream tasks. However, directly applying the parameter-efficient modules obtained from the small language model to the large language model would face two issues: dimension mismatch and limited interaction with the large language model. To address the issue of dimension mismatch, we employ a linear projection module $W_{plug}$ as the Plug-in model to align the dimensions of the parameter-efficient module with the large language model. Furthermore, considering that small language models usually have fewer layers than large language models, we address the layer mismatch by duplicating the layers of the parameter-efficient modules. This duplication enables us to achieve layer alignment with the large language model.

Intuitively, successfully adapting the parameter-efficient modules to the large language models requires a substantial interaction between them and the large language models. To enrich the interaction, we introduce a Bridge model that can retain the knowledge of the large language model while interacting with the parameter-efficient module. We employ the pruning method from Ladder-side-tuning (Sung et al., 2022) to obtain such a Bridge model, which involves pruning each layer of the large language model $f$. We use linear projections to downsample the intermediate activa-

tions, including word embeddings, from the large language model $f$ to a lower-dimensional Bridge model $g$, with a reduction factor of $r$, where $r$ can be 8, 16, 32, 64, etc. To retain crucial information from the large language model, we leverage Fisher information (Liu et al., 2021a; Sung et al., 2021) to prune the parameters of the large language model and obtain the initial Bridge model $g$. Fisher information could effectively evaluate the importance of parameters in the large language model. Given the $\boldsymbol{W} \in \mathbb{R}^{d_b \times d_l}$ of the backbone network that maps the $d_l$-dim vectors to the $d_b$-dim space, we calculate the importance of each weight vector through

$$\boldsymbol{W} = \frac{1}{|D|} \sum_{i=1}^{|D|} (\nabla_W \log p(y_i|x_i))^2,$$

where $(x_i, y_i)$ are samples from data $D$. Then, we keep the rows and columns of the $W$, which have the $\frac{d_b}{r}$ and $\frac{d_l}{r}$ importance scores. Through iterations of this process in each layer of the Transformer, we obtain a set of weight matrices $W_B \in \mathbb{R}^{\frac{d_b}{r} \times \frac{d_l}{r}}$ that have undergone pruning $1/r$ times from the backbone network, and we utilize them to initialize the Bridge model.

Subsequently, in order to make the parameter-efficient modules (e.g., Adapter, LoRA, or Prefix tuning) learned in the small language model interact with the Bridge model, we apply the linear

---

**Algorithm 1** Adaptation of Parameter-efficient Modules for Large Language Model

---

**Require:** Large language model $M$, a small language model $S$, the training set $D = \{(x_1, y_1), \cdots, (x_n, y_n)\}$, linear projection modules $W_{plug}$ and $W_{down}$, and parameter-efficient tuning methods (PEFT) (e.g., Adapter tuning, Prefix tuning, and LoRA)

1: Apply the PEFT on $S$ and fine-tuning $S$ on $D$ to obtain parameter-efficient modules
2: Employ the $W_{plug}$ to align the dimensions of the parameter-efficient modules with $M$
3: Prune the parameters of the $M$ and get a Bridge model $g$
4: Apply the linear projection module $W_{down}$ on parameter-efficient modules
5: **for** each instance $(x_1, y_1)$ in $D$ **do**
6:     Fine-tune the parameter-efficient modules together with the Bridge model $g$
7: **end for**
8: Plug the parameter-efficient modules and the linear projection modules $W_{plug}$ into the $M$

---

projection module $W_{down}$ on parameter-efficient modules and fine-tune the parameter-efficient modules together with the Bridge model $g$. This fine-tuning process enables us to achieve two objectives: obtaining parameter-efficient modules that match the dimensions of the large language model and enriching these modules with knowledge from the large language model.

**Inference:** Once the training of the parameter-efficient modules and the Bridge model $g$ is complete, we integrate the trained parameter-efficient modules, enriched with knowledge from the large language model, into the large language model. This integration empowers the large language model to leverage the task-specific knowledge captured by the parameter-efficient modules during the inference process without requiring gradient-based optimization. The complete algorithm is depicted in Algorithm 1.

## 3 Experiments

### 3.1 Experimental Settings

We conduct extensive experiments on eight natural language understanding tasks from the SuperGLUE benchmark, including BoolQ (Clark et al., 2019), CB (De Marneffe et al., 2019), COPA (Roemmele et al., 2011), MultiRC (Khashabi et al., 2018), RTE (Bar-Haim et al., 2014), WiC (Pilehvar and Camacho-Collados, 2019), WSC (Levesque et al., 2012), and ReCoRD (Zhang et al., 2018). For each task, we report the accuracy or F1-score. In our experiments, we evaluate the effectiveness of our method using both the GPT2 series (Radford et al., 2019) of autoregressive language models and the T5 series (Raffel et al., 2020) of sequence-to-sequence language models. In the GPT2 series of models, we designate the GPT2-base as the small

model and the GPT2-XL as the large model. For the T5 series of models, we classify the T5-base and T5-large as small language models, while the T5-3B and T5-XXL are considered large language models for our experiments. To obtain the Bridge model, we set the reduction factor $r = 16$. In Table 2, we provide the model parameters of the small language models, large language models, and Bridge models. It is worth noting that the model parameters of the Bridge models are significantly smaller than those of the large language models. The training process of our proposed method is conducted on an NVIDIA A100 GPU with 80GB of memory.

Our objective is to demonstrate that the parameter-efficient modules learned on the small language models can be effectively adapted to the large language models, achieving comparable performance to full fine-tuning and parameter-efficient tuning without needing gradient-based optimization. We list the baselines as follows:

**Fine-tuning:** The vanilla Transformer fine-tuning.

**Adapter tuning:** Inserting a small task-specific module between the self-attention module (and the MLP module) and the subsequent residual connection at each Transformer layer (Houlsby et al., 2019).

**Prefix tuning:** Adding trainable continuous prompt vectors to the Key and Value components of the attention layer at each layer of the Transformer model (Li and Liang, 2021).

**LoRA:** Merging the low-rank and trainable matrices with the frozen weights at each layer of the Transformer (Hu et al., 2021).

**Adapter tuning Plug-in:** By applying the Adapter tuning method to small language models,

| Method | #Params | BoolQ | CB | COPA | MultiRC | RTE | WiC | WSC | ReCoRD |
| | | Acc. | Acc./F1 | Acc. | EM/F1a | Acc. | Acc. | Acc. | Acc./F1 |
|---|---|---|---|---|---|---|---|---|---|
| GPT2-base FT (Radford et al., 2019) | 100% | 71.2 | 78.6/55.8 | 64.4 | 65.8/17.4 | 67.8 | 65.5 | 63 | 72.1/71.4 |
| GPT2-base Adapter | - | 71.5 | 79.3/56.3 | 65.8 | 65.2/18.2 | 67.5 | 65.7 | 62.9 | 72.8/71.9 |
| GPT2-base Prefix tuning | - | 70.6 | 80.3/58.2 | 65.1 | 64.7/16.9 | 67.3 | 64.9 | 63.1 | 71.4/70.5 |
| GPT2-base LoRA | - | 71.4 | 79.6/57.8 | 65.7 | 66.2/18.6 | 67.2 | 65.3 | 62.5 | 71.8/70.9 |
| GPT2-XL FT (Radford et al., 2019) | 100% | 82.4 | 87.4/90.6 | 76.5 | 76.4/37.6 | 79.6 | 74.9 | 81.8 | 84.4/83.8 |
| GPT2-XL Adapter | 1.98% | 81.8 | 86.9/90.8 | 76.2 | 75.4/36.2 | 78.9 | 74.2 | 81.5 | 84.2/83.1 |
| GPT2-XL Prefix tuning | 1.76% | 81.0 | 86.7/89.9 | 74.9 | 76.7/36.9 | 79.2 | 74.3 | 81.4 | 83.2/83.1 |
| GPT2-XL LoRA | 1.55% | 82.1 | 87.1/90.3 | 76.2 | 76.2/37.1 | 79.4 | 75.1 | 81.1 | 84.4/83.3 |
| GPT2-XL Adapter Plug-in | 0% | 81.6 | 86.7/90.4 | 75.6 | 75.8/36.7 | 78.7 | 74.3 | 80.8 | 83.7/82.5 |
| GPT2-XL Prefix tuning Plug-in | 0% | 81.2 | 86.3/89.4 | 75.1 | 76.1/36.4 | 78.5 | 73.8 | 80.1 | 83.5/82.6 |
| GPT2-XL LoRA Plug-in | 0% | 81.5 | 86.2/89.3 | 75.3 | 75.6/36.8 | 78.3 | 74.1 | 80.3 | 83.9/82.8 |

Table 1: Results on the SuperGLUE benchmark. The Adapter Plug-in, Prefix tuning Plug-in, and LoRA Plug-in are the parameter-efficient modules learned in the small language models adapted to the large language models. 0% means we do not update any parameter within the large language models.

we obtain parameter-efficient modules that can be adapted to large language models.

**Prefix tuning Plug-in:** By applying the Prefix tuning method to small language models, we obtain parameter-efficient modules that can be adapted to large language models.

**LoRA Plug-in:** By applying the LoRA method to small language models, we obtain parameter-efficient modules that can be adapted to large language models.

| Models | SLMs | LLMs | BMs |
|---|---|---|---|
| GPT2-base and GPT2-XL | 117M | 1542M | 96M |
| T5-base and T5-3B | 220M | 2800M | 175M |
| T5-large and T5-XXL | 770M | 11000M | 688M |

Table 2: Comparison of the model parameters on the small language models (SLMs), large language models (LLMs), and the Bridge models (BMs). To obtain the Bridge model, we set the reduction factor $r = 16$.

## 3.2 Main Results

### 3.2.1 Experiments on GPT-2 Series of Models

Table 1 shows the performance of our proposed method using the GPT2 series of models (Radford et al., 2019). In Table 1, we present the results obtained with GPT2-base as the small language model and GPT2-XL as the large language model. The Adapter Plug-in, Prefix tuning Plug-in, and LoRA Plug-in are the parameter-efficient modules learned in the GPT2-base model adapted to the GPT2-XL model. As can be seen, in comparison to directly conducting vanilla fine-tuning on the large language model, our method achieves comparable results. Furthermore, compared to

parameter-efficient tuning methods applied to the large language model, our method demonstrates comparable performance without the need to fine-tune any parameters of the large language model. Specifically, it exhibits a slight improvement compared to the prefix-tuning method on the BoolQ and COPA tasks, demonstrating the effectiveness of our method.

### 3.2.2 Experiments on T5 Series of Models

Table 3 and Table 4 display the performance of our proposed method using the T5 series of models (Raffel et al., 2020). In Table 3, we present the results obtained with T5-base as the small language model and T5-3B as the large language model. The Adapter Plug-in, Prefix tuning Plug-in, and LoRA Plug-in are parameter-efficient modules of our method used to adapt T5-3B. Our method achieves comparable results on all eight Super-GLUE tasks without the need to fine-tune any parameters of the T5-3B, demonstrating the effectiveness of our method. Particularly in BoolQ, CB, RTE, and ReCoRD, our method performs at a level similar to full fine-tuning. Similarly, Table 4 shows the results using T5-large as the small language model and T5-XXL as the large language model, with the same Plug-ins employed to adapt T5-XXL. Our method achieves comparable results on all eight SuperGLUE tasks without the need to fine-tune any parameters of the T5-XXL. Furthermore, compared to parameter-efficient tuning methods applied to the large language model, our method demonstrates comparable performance without the need to fine-tune any parameters of the large language model. Especially when compared to the

| Method | #Params | BoolQ Acc. | CB Acc./F1 | COPA Acc. | MultiRC EM/F1a | RTE Acc. | WiC Acc. | WSC Acc. | ReCoRD Acc./F1 |
|---|---|---|---|---|---|---|---|---|---|
| T5-base FT (Raffel et al., 2020) | 100% | 81.4 | 81.4/91.0 | 71.2 | 79.7/43.1 | 81.5 | 68.3 | 80.8 | 75.0/74.2 |
| T5-base Adapter | - | 81.5 | 82.6/93.5 | 71.5 | 79.3/43.2 | 81.2 | 68.8 | 80.3 | 75.2/74.6 |
| T5-base Prefix tuning | - | 81.1 | 81.9/93.2 | 70.1 | 79,2/42.1 | 80.8 | 67.9 | 79.9 | 74.6/73.5 |
| T5-base LoRA | - | 81.7 | 82.1/94.3 | 70.4 | 79.5/42.6 | 81.2 | 69.1 | 80.5 | 74.8/74.1 |
| T5-3B FT (Raffel et al., 2020) | 100% | 89.9 | 90.3/94.4 | 92 | 86.8/58.3 | 90.7 | 72.1 | 90.4 | 91.2/90.4 |
| T5-3B Adapter | 2.73% | 89.7 | 90.1/93.2 | 91.5 | 86.9/59.1 | 91.1 | 71.3 | 90.5 | 91.1/90.3 |
| T5-3B Prefix tuning | 2.12% | 89.4 | 89.0/93.4 | 91.1 | 86.4/57.1 | 90.5 | 70.9 | 89.1 | 90.6/89.2 |
| T5-3B LoRA | 2.45% | 90.4 | 90.8/95.0 | 92.1 | 86.2/57.6 | 91.0 | 71.8 | 90.6 | 91.3/91.1 |
| T5-3B Adapter Plug-in | 0% | 89.5 | 89.6/94.7 | 91.7 | 86.5/58.2 | 90.8 | 70.9 | 90.2 | 91.6/90.7 |
| T5-3B Prefix tuning Plug-in | 0% | 89.1 | 89.4/93.8 | 91.2 | 86.1/57.4 | 90.4 | 70.7 | 89.5 | 90.3/89.9 |
| T5-3B LoRA Plug-in | 0% | 89.6 | 90.0/94.9 | 91.8 | 86.9/58.5 | 90.1 | 71.0 | 89.8 | 90.9/90.1 |

Table 3: Results on the SuperGLUE benchmark. The Adapter Plug-in, Prefix tuning Plug-in, and LoRA Plug-in are the parameter-efficient modules learned in the small language models adapted to the large language models. The full fine-tuning performance of T5-base and T5-3B models is based on the research conducted by Raffel et al. (2020). 0% means we do not update any parameter within the large language models.

| Method | #Params | BoolQ Acc. | CB Acc./F1 | COPA Acc. | MultiRC EM/F1a | RTE Acc. | WiC Acc. | WSC Acc. | ReCoRD Acc./F1 |
|---|---|---|---|---|---|---|---|---|---|
| T5-large FT (Raffel et al., 2020) | 100% | 85.4 | 91.6/94.8 | 83.4 | 83.3/50.7 | 87.8 | 69.3 | 86.3 | 86.8/85.9 |
| T5-large Adapter | - | 85.3 | 90.9/93.7 | 83.2 | 83.1/50.1 | 87.3 | 69.8 | 86.7 | 86.2/85.1 |
| T5-large Prefix tuning | - | 84.9 | 91.4/95.3 | 82.9 | 82.7/49.6 | 87.5 | 68.9 | 85.5 | 86.1/84.9 |
| T5-large LoRA | - | 85.1 | 90.9/94.3 | 84.2 | 83.6/51.2 | 87.3 | 70.2 | 85.9 | 86.7/85.7 |
| T5-XXL FT (Raffel et al., 2020) | 100% | 91.2 | 93.9/96.8 | 94.8 | 88.1/63.3 | 92.5 | 76.9 | 93.8 | 94.1/93.4 |
| T5-XXL Adapter | 6.42% | 90.8 | 93.6/95.9 | 93.7 | 87.9/63.0 | 92.8 | 75.7 | 93.5 | 92.8/91.7 |
| T5-XXL Prefix tuning | 6.22% | 89.3 | 93.5/94.4 | 92.7 | 86.9/63.1 | 92.3 | 74.9 | 92.8 | 93.1/91.8 |
| T5-XXL LoRA | 6.15% | 91.5 | 93.4//95.8 | 93.6 | 87.8/63.5 | 92.7 | 75.7 | 93.6 | 93.7/92.8 |
| T5-XXL Adapter Plug-in | 0% | 90.4 | 93.1/95.6 | 94.1 | 87.2/62.8 | 92.3 | 75.2 | 93.2 | 93.4/92.2 |
| T5-XXL Prefix tuning Plug-in | 0% | 89.9 | 93.2/94.6 | 92.3 | 86.3/62.1 | 91.8 | 74.7 | 92.4 | 93.5/92.5 |
| T5-XXL LoRA Plug-in | 0% | 90.5 | 92.4//95.7 | 93.2 | 86.9/62.5 | 92.1 | 74.9 | 92.9 | 93.8/92.7 |

Table 4: Results on the SuperGLUE benchmark. The Adapter Plug-in, Prefix tuning Plug-in, and LoRA Plug-in are the parameter-efficient modules learned in the small language models adapted to the large language models. The full fine-tuning performance of T5-large and T5-XXL models is based on the research conducted by Raffel et al. (2020). 0% means we do not update any parameter within the large language models.

prefix-tuning method, our proposed method shows a slight improvement in the CB, COPA, and WSC tasks when applied to the T5-3B model. Similarly, when applied to the T5-XXL model, our method slightly improves the BoolQ, CB, and ReCoRD tasks.

It is evident that as the parameter size of pre-trained language models grows, existing parameter-efficient tuning methods necessitate the addition of more parameters to the large language model. In contrast, our method does not entail any parameter augmentation but instead optimally harnesses the capabilities of the large language model through a plug-in approach. These findings indicate that our method can effectively utilize the knowledge of large language models without updating the pa-

rameters of large language models, which suggests the potential for application to even larger language models.

### 3.2.3 Importance of the Reduction Factor $r$

Considering the impact of the reduction factor $r$ on the amount of knowledge retained in the Bridge model, we conduct experiments to analyze its importance. For verification, we select CB, RTE, and WiC in the SuperGLUE benchmark. As indicated in Table 5, we observe a gradual decrease in model performance as $r$ increases. This is because a higher value of $r$ reduces the retained knowledge from the large language model to the Bridge model. According to Table 5, we can observe that when the model parameters of the Bridge model

| Dataset | r=2 | r=4 | r=8 | r=16 | r=32 | r=64 |
|---|---|---|---|---|---|---|
| CB | 91.2 | 90.6 | 90.2 | 90.0 | 88.2 | 85.4 |
| RTE | 90.8 | 90.5 | 90.5 | 90.1 | 89.1 | 84.3 |
| WiC | 71.3 | 71.5 | 71.1 | 71.0 | 69.7 | 67.4 |

Table 5: The accuracy on CB, RTE, and WiC with different reduction factors $r$. We perform the verification on T5-3B using the LoRA Plug-in approach.

are significantly smaller than that of the small language model, our method exhibits a noticeable performance decline during the entire inference process. However, when the model parameters of the Bridge model are comparable to that of the small language model, our method maintains the performance of the model without substantial degradation. To strike a balance between model performance and the parameter of the Bridge model, we believe that selecting $r = 16$ is a suitable choice.

### 3.2.4 Memory Usage

Our method not only achieves comparable performance to full fine-tuning and parameter-efficient tuning methods without updating any parameters of the large language models but also significantly achieves an impressive reduction in memory usage. As shown in Table 6 and Table 7, we compare the memory usage between our proposed method and the baseline models. When using the GPT2-XL model, we conduct experiments with a batch size of 8 and a sequence length 512. Our method evidently achieves up to 7.1× memory savings compared to vanilla fine-tuning. Similarly, when utilizing the T5-3B model with a batch size of 1 and a sequence length 512, our method achieves up to 5.1× memory savings compared to vanilla fine-tuning.

In particular, we compare the existing parameter-efficient tuning methods, and in GPT2-XL, we can see that our proposed method can achieve significant memory savings. For example, comparing the three parameter-efficient tuning methods of Adapter-tuning, Prefix tuning, and LoRA, our proposed method can achieve 5.3×, 5.6×, and 5.7× memory reduction, respectively. Our method also achieves 2.9× more memory savings compared to Ladder-side Tuning. In T5-3B, our proposed method can achieve 3.6×, 3.5×, and 3.6× memory reduction, respectively, compared to the Adapter-tuning, Prefix tuning, and LoRA. Similarly, our method also achieves 1.3× more memory savings compared to Ladder-side Tuning. This demonstrates that our method can be more effectively

applied to existing large language models without compromising performance. Furthermore, our proposed method does not slow down the inference speed of the model. By utilizing the plug-in approach, we can directly leverage the knowledge of the large language model during inference without compromising the speed.

| Models | MU | MR |
|---|---|---|
| GPT2-XL FT | 73746 | 1.0× |
| GPT2-XL Adapter | 57903 | 1.3× |
| GPT2-XL Prefix tuning | 56051 | 1.3× |
| GPT2-XL LoRA | 54593 | 1.4× |
| Ladder-side Tuning | 17652 | 4.2× |
| GPT2-XL Adapter Plug-in | 11203 | 6.6× |
| GPT2-XL Prefix tuning Plug-in | 10745 | 6.9× |
| GPT2-XL LoRA Plug-in | 10443 | **7.1×** |

Table 6: Memory Usage (MU) and Memory Reduction (MR) compared to vanilla fine-tuning of our proposed method on a single NVIDIA A100 GPU with 80GB of memory. Batch sizes are 8 and sequence lengths are 512.

| Models | MU | MR |
|---|---|---|
| T5-3B FT | 77465 | 1.0 × |
| T5-3B Adapter | 54324 | 1.4× |
| T5-3B Prefix tuning | 53324 | 1.5× |
| T5-3B LoRA | 52134 | 1.5× |
| Ladder-side Tuning | 20385 | 3.8× |
| T5-3B Adapter Plug-in | 15432 | 5.0× |
| T5-3B Prefix tuning Plug-in | 15643 | 5.0× |
| T5-3B LoRA Plug-in | 15323 | **5.1×** |

Table 7: Memory Usage (MU) and Memory Reduction (MR) compared to vanilla fine-tuning of our proposed method on a single NVIDIA A100 GPU with 80GB of memory. Batch sizes are 1 and sequence lengths are 512.

### 3.3 Utilize Bridge Model Directly?

A natural question arises: Why not learn the parameter-efficient modules directly on the Bridge model instead of using a smaller language model and then applying the learned efficient modules to the larger language model? Intuitively, given the learned parameter-efficient modules with the Bridge model, we need a projection model to project the dimensions to match the large language model. However, the projection model can only be well learned with another interaction model. In this section, we train the Bridge model by directly initializing the parameter-efficient and lin-

| Method | COPA | RTE | WiC | WSC |
|---|---|---|---|---|
| Adapter Plug-in (BM) | 90.3 | 89.9 | 69.4 | 89.3 |
| LoRA Plug-in (BM) | 90.7 | 89.8 | 70.1 | 89.4 |
| Adapter Plug-in (SLM) | 91.7 | 90.8 | 70.9 | 90.2 |
| LoRA Plug-in (SLM) | 91.8 | 90.1 | 71.0 | 89.8 |

Table 8: The Accuracy on COPA, RTE, WiC, and WSC tasks with T5-3B. BM means the Bridge model obtained with T5-3B. SLM means the small language model T5-base.

ear projection modules and investigate whether a Bridge model is enough to learn the projection module. In Table 8, we conduct experiments using the T5-3B model on the COPA, RTE, WiC, and WSC tasks. By plugging the parameter-efficient modules learned from the Bridge model and the small language model into the T5-3B model, respectively, and comparing their performance, we find that directly using the Bridge model consistently performs worse than utilizing the small language model enriched by a Bridge model across all tasks. The results indicate that the Bridge model cannot be utilized alone.

## 4 Related Work

**Fine-tuning and Parameter-efficient Tuning:** Large language models leverage parameterized Transformers as a foundational framework and train them on extensive unsupervised language corpora (Brown et al., 2020; OpenAI, 2023; Touvron et al., 2023; Raffel et al., 2020). Subsequently, specific task objectives are introduced for downstream tasks to perform full fine-tuning on the pre-trained language models (Jin et al., 2022, 2023; Muennighoff et al., 2022). As the primary method for optimizing pre-trained language models, full fine-tuning involves initializing the model with pre-trained weights, updating all model parameters, and storing a separate fully optimized model for each downstream task. However, as the model parameter grows, conducting full fine-tuning on existing computational devices becomes increasingly challenging. In light of this, researchers have started exploring efficient methods for effectively harnessing the power of large language models.

As an efficient alternative, parameter-efficient tuning is a promising way of stimulating large language models. Compared to vanilla fine-tuning, parameter-efficient tuning methods only tune a small portion of the model parameters while keeping the rest frozen. Adapter tuning methods,

such as those proposed by Houlsby et al. (2019) and Mahabadi et al. (2021b,a), aim to learn task-specific information by incorporating small-scale task-specific modules into the layers of the Transformer. Prefix tuning methods, as introduced by Li and Liang (2021) and discussed in the work by Liu et al. (2021b), also introduce additional parameters within the Transformer layers. LoRA, proposed by Hu et al. (2021), merges low-rank and trainable matrices with the frozen weights at each layer of the Transformer. BitFit (Ben Zaken et al., 2022) is a simple yet effective method that optimizes the bias terms within a model while keeping the other parameters frozen. However, these existing parameter-efficient tuning methods typically rely on gradient-based optimization and still involve substantial memory usage. Our proposed method, on the other hand, enables substantial memory savings while maintaining comparable performance.

**Gradient-free Optimization:** Recently, Sung et al. (2022) introduced a method that eliminates the need for gradient updates by directly applying a pruned model to downstream tasks, while the method does not fully exploit the knowledge of the large language model. Xiao et al. (2023) propose an efficient transfer learning framework that can adapt large language models to downstream tasks without access to full model parameters, while the method through compute-intensive distillation techniques may be cost-prohibitive for larger models. In contrast, our proposed method enables further utilization of the knowledge contained in the large language model and enables significant memory savings through a simple operation while preserving the performance of large language models.

## 5 Conclusion

This paper proposes a novel parameter-efficient tuning method for large language models without calculating their gradients. We first learn the parameter-efficient tuning module for small language models. Then, the learned parameter-efficient tuning module is adapted into large language models with a bridge model that handles the dimensionality mismatch and enables interaction between the parameter-efficient tuning module and the large language model. Extensive experiments on the SuperGLUE benchmark demonstrate that our method achieves comparable performance to vanilla fine-tuning and parameter-efficient tuning on large language models without needing gradient-

based optimization. We believe that our method offers a potential direction to utilize large language models efficiently and economically.

## Limitations

For large language models for which weights cannot be obtained, the proposed methods cannot be directly applied, and there may be certain limitations when applying the proposed methods to language models with different architectures. Continuous exploration and research will be conducted to determine how to apply the proposed methods to different architecture language models, aiming to improve their compatibility and effectiveness across different architectures and language model characteristics.

The proposed method still updates the parameters of PEFT modules based on the gradient on a small language model, and the whole process is a pipeline process, which requires practical training in the early stage before it can be applied to the large model.

## Acknowledgement

This work is supported by National Key R&D Program of China 2022ZD0160602 and the Natural Science Foundation of China 62122088.

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
