# OpenReview forum: "Parameter-efficient Tuning for Large Language Model without Calculating Its Gradients"
_EMNLP/2023/Conference — EMNLP 2023 Main_

### Official Review · Reviewer_ZcTW · 2023-08-02

**Soundness:** 3

**Excitement:**

3: Ambivalent: It has merits (e.g., it reports state-of-the-art results, the idea is nice), but there are key weaknesses (e.g., it describes incremental work), and it can significantly benefit from another round of revision. However, I won't object to accepting it if my co-reviewers champion it.

**Missing References:**

Sun, T., Shao, Y., Qian, H., Huang, X., & Qiu, X. (2022). Black-Box Tuning for Language-Model-as-a-Service (arXiv:2201.03514).

Sun, T., He, Z., Qian, H., Zhou, Y., Huang, X., & Qiu, X. (2022). BBTv2: Towards a Gradient-Free Future with Large Language Models

Hou, B., O’Connor, J., Andreas, J., Chang, S., & Zhang, Y. (2022). PromptBoosting: Black-Box Text Classification with Ten Forward Passes (arXiv:2212.09257).

Malladi, S., Gao, T., Nichani, E., Damian, A., Lee, J. D., Chen, D., & Arora, S. (2023). Fine-Tuning Language Models with Just Forward Passes (arXiv:2305.17333).

**Paper Topic And Main Contributions:**

This work presents a memory-efficient tuning pipeline by first training the PEFT module on SML. With an interaction of a Bridge model that is pruned from the LLM, this work further trains the PEFT module with the bridge model. It also employs a linear projection layer to overcome the dimensionality mismatch between the SLM and LLM. It shows consistent results with up to 5 times memory footprint reduction on T5 and GPT-2 models. However, this work only considers baselines with naive PEFT modules, where the comparison to important baselines like LST, BBT, BBTv2 is missing. This raises my concerns about the solidness of this work.

Sung, Y.-L., Cho, J., & Bansal, M. (2022). LST: Ladder Side-Tuning for Parameter and Memory Efficient Transfer Learning.

**Questions For The Authors:**

1. Can you provide the mentioned baseline performance in the weakness section?

2. Please explain why '0% #params' is indicated here.

If the author could address my concerns, I will increase my scores.

### After Rebuttal

Thanks the author for providing additional evidence for a comparison to the LST baseline. I have increased the soundness score.

**Reasons To Accept:**

1. Transferring the PEFT learned from the SLM to LLM is a great idea due to the cost of FT LLM.
2. This work shows significantly reduced memory usage and reasonable performance degradation.
3. It conducts experiments on two types of backbones with convincing results.

**Reasons To Reject:**

1. Important baselines are missing. The author did cite LST but there is no comparison between the performance of LST and the proposed method, where LST also shows significantly reduced memory but competitive performance to FFT. It is important that other black-box forward-only (without calculating gradients) baselines that also reduce memory costs are missing, including but not limited to BBT, BBTv2, MeZo, PromptBoosting.
2. '0% #params' descriptions in Table 1, 3, 4 are misleading. The author should clearly state the total trainable parameters and the parameter required to be saved during the pipeline instead of stating 0%.
3. The description in Lines 396-398 is incorrect. Only LoRA Plug-in does not slow the inference speed.

**Reproducibility:**

4: Could mostly reproduce the results, but there may be some variation because of sample variance or minor variations in their interpretation of the protocol or method.

**Reviewer Confidence:**

4: Quite sure. I tried to check the important points carefully. It's unlikely, though conceivable, that I missed something that should affect my ratings.

---

> ### Author Rebuttal · Authors · 2023-08-25
>
> Thank you for your comments. We address your concerns one by one as follows.
> Q1: Can you provide the mentioned baseline performance in the weakness section?
> In the T5-3B model, we observe that LST achieves 2.4× more memory savings compared to Adapter and LoRA, in line with the findings of the original paper. Moreover, our method outperforms LST with a substantial memory saving of 3.5× and 3.6× over Adapter and LoRA, respectively. When analyzing the BBT and BBTv2 methods, their implementation like prefix-tuning on GPT2-XL and T5-3B models introduces additional parameters, leading to heightened memory consumption and diminished inference speed. For instance, in the T5-3B model, BBT incurs a memory usage of approximately 19473MB, whereas BBTv2 utilizes around 22749MB. In contrast, our proposed method showcases a memory requirement of approximately 15643MB, affirming its efficiency in memory management.
> Q2: Please explain why '0% #params' is indicated here.
> '0% #params' is expected to show that we do not update any parameter within the larger model. We will also provide accurate and transparent information about the total trainable parameters and the specific parameters that are preserved during the pipeline, rather than using only '0%'.
> Q3: The description in Lines 396-398 is incorrect. Only the LoRA Plug-in does not slow the inference speed.
> Thanks for your reminding. We will refine the corresponding wording in the revised version.

---

### Official Review · Reviewer_wTP7 · 2023-08-04

**Soundness:** 4

**Excitement:**

4: Strong: This paper deepens the understanding of some phenomenon or lowers the barriers to an existing research direction.

**Paper Topic And Main Contributions:**

This paper provides a novel parameter efficient tuning method for large language models.
This method learns a parameter efficient module of a small language model, and then adapt this module to the large language model using a linear projection module and a bridge model.
The experiment shows that this method achieves significant memory reduction during training, and achieves comparable performances in various tasks.

**Reasons To Accept:**

1. Excessive memory usage is a typical problem during fine-tuning or parameter-efficient tuning. It is an interesting idea to combine both SLM and LLM, and just calculate the gradient of SLM during training.

2. The proposed method achieves comparable performance while significantly reduces memory usage, and may promote the application of large language models in downstream tasks.

3. Solid experiments and detailed analysis are carried out in the paper.

**Reasons To Reject:**

The pictures and tables in the article can be further refined.

1. The legend in Figure 1 could be reordered so that it appears in the same order as it appears in the figure.

2. The experimental results of the proposed method in Tables 1, 3, and 4 could be highlighted, for example, the results that are less than 1 point lower than the corresponding parameter-efficient methods could be underlined.

**Reproducibility:**

4: Could mostly reproduce the results, but there may be some variation because of sample variance or minor variations in their interpretation of the protocol or method.

**Reviewer Confidence:**

2: Willing to defend my evaluation, but it is fairly likely that I missed some details, didn't understand some central points, or can't be sure about the novelty of the work.

---

> ### Author Rebuttal · Authors · 2023-08-25
>
> Thank you for your encouraging comments. We address your concerns as follows.
> Q1: The pictures and tables in the article can be further refined.
> Thanks for your suggestion. In the revision, we will make the pictures and tables much more clear.

---

### Official Review · Reviewer_djcv · 2023-08-04

**Typos Grammar Style And Presentation Improvements:** 1. Alrogrithm 1
**Soundness:** 3

**Excitement:**

2: Mediocre: This paper makes marginal contributions (vs non-contemporaneous work), so I would rather not see it in the conference.

**Missing References:**

In related work, it's better to summarize some related works to rescue training memory, like:

[1] Dettmers, Tim, et al. "Qlora: Efficient finetuning of quantized llms."

[2] Liu, Yitao, Chenxin An, and Xipeng Qiu. "$\mathcal {Y} $-Tuning: An Efficient Tuning Paradigm for Large-Scale Pre-Trained Models via Label Representation Learning.

[3] Liao, Baohao, Shaomu Tan, and Christof Monz. "Make Your Pre-trained Model Reversible: From Parameter to Memory Efficient Fine-Tuning."

**Paper Topic And Main Contributions:**

This paper is about reducing the memory footprint when fine-tuning a large language model. The authors propose to fine-tune the parameter-efficient fine-tuning (PEFT) modules on a smaller pre-trained model, then project these modules to the same dimension of the larger language model by further training the projection matrix on a pruned variant of the larger language model and finally plug these leaned modules to the larger language model.

The main contributions of this paper are:
(1) The proposed method reduces the memory footprint significantly, while slightly sacrificing some scores.
(2) Extensive experiments are conducted on two series of models (GPT and T5) and SuperGLUE tasks.

**Questions For The Authors:**

1. When do you train W_{plug}  and W_{down}? Also in Line 6 of Algorithm 1?

2. Line 294~297: This claim is not true. Out of 24 experiments (3 PEFT methods for 8 tasks), only two experiments are better.


**Reasons To Accept:**

1. The proposed method could achieve comparable results as the one directly fine-tuning the larger language model, saving a large amount of memory footprint.

2. Extensive experiments are conducted to support the main claim.

**Reasons To Reject:**

1. The title and the way to show results are misleading:
(1) The proposed method still updates the parameters of PEFT modules based on the gradient. Instead of fine-tuning the PEFT module directly on the larger language model, the proposed method fine-tunes the PEFT module on a smaller model and applies it directly to the larger language model without further training. But it doesn't mean that it's gradient-free.
(2) In all tables with #Params, the authors use 0% for the proposed methods. In all related works, #Params means the number of trainable/new parameters. It's not 0% for your work in this case. If you want to highlight that you don't need to further train the PEFT modules when plugging into the larger model, you need to specify it rather than use it in this way.
(3) In all performance result tables (Table 1, 3, 4), the authors include the results from the smaller language model. However, when comparing the memory footprint for training, all baselines for the smaller language model are excluded.

2. Experimental settings are missing, like the grid search space for learning rate, batch size and etc. All results are shown without an error bar. SuperGLUE tasks are very sensitive to hyper-parameter settings. It's better to show the error bar (like variance) for the main results.

3. The writing is not easy to follow. And the contribution is limited, mainly combining PEFT and pruning methods together to reduce the memory footprint for training.

**Reproducibility:**

3: Could reproduce the results with some difficulty. The settings of parameters are underspecified or subjectively determined; the training/evaluation data are not widely available.

**Reviewer Confidence:**

4: Quite sure. I tried to check the important points carefully. It's unlikely, though conceivable, that I missed something that should affect my ratings.

---

> ### Author Rebuttal · Authors · 2023-08-25
>
> Thank you for your comments. We address your concerns one by one as follows.
> Q1: The title and the way to show results are misleading...，
> (1) About the title, the fact is that our proposed method fine-tunes the PEFT modules on a much smaller model before applying it to the larger model, without further training the large model. That is we did not update any parameters on the large model. Hence, the title is expected to convey the meaning that no gradient updates are conducted on the large model.
> (2) The utilization of 0% for #Params indeed seems misleading. We just want to say that the incorporation of PEFT modules into the larger model does not introduce any additional trainable parameters for the larger model. We will follow your suggestion to use another term.
> (3) As the purpose of the small model is to learn parameter-efficient modules tailored for the large model, our primary concern centers around optimizing memory efficiency while maintaining strong performance in the large model. Since our method aims to enhance the utilization of the large model, we focus more on comparing the memory usage of the large model.
> Q2: All results are shown without an error bar. SuperGLUE tasks are very sensitive to hyper-parameter settings. It's better to show the error bar (like variance) for the main results...
> Thanks for your reminding. We notice similar variance and we will supplement the information in the revised version.
> Q3: The contribution is limited, mainly combining PEFT and pruning methods together to reduce the memory footprint for training...
> As for the contributions, we believe our method will much benefit the parameter-efficient tuning for very large models. Specifically, we first reveal a substantial task similarity when applying parameter-efficient tuning methods to both SLMs and LLMs for downstream tasks. Then we train the PEFT module with the bridge model and employ a linear projection layer to overcome the dimensionality mismatch between the SLM and LLM. Finally, our proposed method achieves up to 5.7× memory reduction compared to parameter-efficient tuning. We believe that the speedup is promising.
> Q4: When do you train W_{plug} and W_{down}? Also in Line 6 of Algorithm 1?
> As shown in Algorithm 1, we train W_{plug} and W_{down} with the bridge model together.
> Q5: Grammar Style, Presentation Improvements, and References.
> We will revise any inappropriate phrasing and address usage issues accordingly. Additionally, we will incorporate the appropriate references as you suggested.

---

### Meta-Review · Area_Chair_gCM8 · 2023-09-20

**Recommendation:** 4

**Metareview:**

This paper proposes a new parameter- and memory-efficient fine-tuning method that shows significant memory saving compared to other PEFT methods while achieving competitive performance. Concretely, the paper proposes first fine-tuning PEFT modules in a small language model learning task characteristics, then fine-tuning the resulting PEFT models with up/down projections (plug-in module) and a bridge model that is pruned from the large language model. For the inference, the final PEFT modules are inserted into large LM with plug-in up projection without further fine-tuning. The authors presented extensive experiments that include 2 model families (GPT, T5) at different sizes that are evaluated on SuperGLUE benchmark.

As the reviewer mentioned, the proposed method consistently achieves comparable performance with full fine-tuning large model saving a significant amount of memory footprint. Compared to standard PEFTs, the proposed method achieves up to 5.7x memory reduction. The experimental setup is sound and the results are convincing.

However, also the reviewers noted, that phrasing and the number of updated parameters in tables (as 0%) may mislead the conclusion. PEFT modules are updated but using small LMs (and the bridge model) which leads to significant memory reduction. I believe rephrasing or clarification highlighting this point will improve the paper. Connected to this point, the authors acknowledged the issue with the updated parameter number in the tables.

Furthermore, the authors provided an additional comparison with another memory-efficient baseline based on the reviewer's suggestion during the discussion period which fill the corresponding gap.

---

### Decision · Program_Chairs · 2023-10-07

**Decision:**

Accept-Main

**Comment:**

This paper proposes a new parameter- and memory-efficient fine-tuning method that shows significant memory saving compared to other PEFT methods while achieving competitive performance. Concretely, the paper proposes first fine-tuning PEFT modules in a small language model learning task characteristics, then fine-tuning the resulting PEFT models with up/down projections (plug-in module) and a bridge model that is pruned from the large language model. For the inference, the final PEFT modules are inserted into large LM with plug-in up projection without further fine-tuning. The authors presented extensive experiments that include 2 model families (GPT, T5) at different sizes that are evaluated on SuperGLUE benchmark.

As the reviewer mentioned, the proposed method consistently achieves comparable performance with full fine-tuning large model saving a significant amount of memory footprint. Compared to standard PEFTs, the proposed method achieves up to 5.7x memory reduction. The experimental setup is sound and the results are convincing.

However, also the reviewers noted, that phrasing and the number of updated parameters in tables (as 0%) may mislead the conclusion. PEFT modules are updated but using small LMs (and the bridge model) which leads to significant memory reduction. I believe rephrasing or clarification highlighting this point will improve the paper. Connected to this point, the authors acknowledged the issue with the updated parameter number in the tables.

Furthermore, the authors provided an additional comparison with another memory-efficient baseline based on the reviewer's suggestion during the discussion period which fill the corresponding gap.